

# Sandy loam soil maintains better physicochemical parameters and more abundant beneficial microbiomes than clay soil in *Stevia rebaudiana* cultivation

Xinjuan Xu[1,*], Qingyun Luo[2,*], Ningnan Zhang[3], Yingxia Wu[4], Qichao Wei[4], Zhongwen Huang[1] and Caixia Dong[2]

[1] Henan Institute of Science and Technology, School of Agriculture, Collaborative Innovation Center of Modern Biological Breeding, China
[2] College of Horticulture, Nanjing Agricultural University, Nanjing, China
[3] Research Institute of Tropical Forestry, Chinese Academy of Forestry, Guangzhou, China
[4] School of Life Sciences, Henan Institute of Science and Technology, Xinxiang, China
[*] These authors contributed equally to this work.

Corresponding authors
Xinjuan Xu, xuxinjuan@hist.edu.cn
Qingyun Luo, qyluo@njau.edu.cn

## ABSTRACT

Depending on the texture of soil, different physicochemical and microbiological parameters are characterized, and these characteristics are influenced by crop cultivation. *Stevia*, a popular zero-calorie sweetener crop, is widely cultivated around the world on various soil textures. Sandy loam and clay soil show great differences in physicochemical and biological parameters and are often used for *Stevia* cultivation. To understand the effects of *Stevia* cultivation on soil physicochemical and biological features, we investigated the changes of physicochemical and microbiological parameters in sandy loam and clay soil following *Stevia* cultivation. This study was carried out through different physiological and biochemical assays and microbiomic analysis. The results indicated that the sandy loam soil had significantly lower pH and higher nutrient content in the rhizosphere and bulk soils after the *Stevia* cultivation. The sandy loam soil maintained higher bacterial diversity and richness than the clay soil after *Stevia* harvest. Beneficial bacteria such as *Dongia, SWB02, Chryseolinea, Bryobacter* and *Devosia* were enriched in the sandy loam soil; however, bacteria such as *RB41, Haliangium* and *Ramlibacter*, which are unfavorable for nutrient accumulation, predominated in clay soil. Redundancy analysis indicated that the variation in the composition of bacterial community was mainly driven by soil pH, organic matter, total nitrogen, available phosphorus, and microbial biomass phosphorus. This study provides a deeper understanding of physicochemical and microbiological changes in different soil textures after *Stevia* cultivation and guidance on fertilizer management for *Stevia* rotational cultivation.

## INTRODUCTION

Different soil textures have distinct physicochemical characteristics, microbial species, and microbial amounts. Clay soils had a greater genetic potential to degrade organic carbon (*Xia, Rufty & Shi, 2020*). Loam soil has higher nitrogen-use efficiency, nitrogen harvest index, and nitrogen physiological utilization rate than clay soil (*Ye et al., 2007*). The abundance of rhizosphere microorganisms and soil enzyme activities usually differ significantly in different textures of soil, such as sandy soil, medium soil, and heavy soil (*Gebauer et al., 2021*). In maize (*Zea mays*) cultivation, loam soil was found to contain a greater abundance of *Tetrasphaera*, *Amycolatopsis*, and *Streptomyces*, but sandy soil has higher amounts of *Burkholderia*, *Paraburkholderia*, and *Variovorax* (*Gebauer et al., 2021*). Different soil textures may also affect the abundance and variety of plant pathogenic microorganisms (*Gill, Sivasithamparam & Smettem, 2000*; *Sheng et al., 2020*).

Soil microbial communities are dependent on the nature of the soil. The structure and diversity of soil bacterial communities are correlated with soil textures, and to some extent, reflect trends in soil quality (*McCaig, Glover & Prosser, 2001*; *Sajjad et al., 2021*). The soil microbiome is involved in the processes of soil mineralization, nitrogen and phosphorus cycling, nutrient uptake, and fertility maintenance, which play an important role in maintaining the stability of the soil ecosystem (*Burns et al., 2015*; *Trap et al., 2015*; *Hermans et al., 2020*). Bacterial communities are critical for the robustness and productivity of soil ecosystems and have become an essential indicator of soil fertility (*Hermans et al., 2017*). The increase of beneficial soil microorganisms can promote crop yield and quality (*Koskey et al., 2021*; *Ortiz & Estibaliz, 2022*). For example, applying beneficial bacteria or other measures to improve soil beneficial bacteria can increase the yield and quality of *Stevia rebaudiana* (*Xu et al., 2013*; *Sun et al., 2022*).

Crop cultivation may alter soil characteristics. Crop cultivation can influence the microflora diversity and other physicochemical parameters of the soil (*Zhang et al., 2023*; *He et al., 2022*). Continuous peanut cultivation increases soil effective potassium, effective phosphorus, effective nitrogen, soil organic matter, urease, acid phosphatase, and catalase activities (*Li et al., 2022*). Corn rotation significantly reduces the contents of manganese, copper, and five phenolic acids in ginseng (*Panax ginseng*) cultivation soil (*Jiao et al., 2019*). Rotation between legumes and other crops often increases soil fertility, raises soil organic matter content, and improves soil microbiomes (*Wang et al., 2023a*; *Wang et al., 2023b*; *MacWilliam, Wismer & Kulshreshtha, 2014*; *Essel et al., 2019*). Long-term continuous cropping of some plants may lead to a decrease in the relative abundance of beneficial microorganisms and an increase in the relative abundance of pathogenic ones (*Zhang et al., 2020a*; *Zhang et al., 2020b*; *Zhang et al., 2022a*; *Zhang et al., 2022b*; *Wang et al., 2023a*; *Wang et al., 2023b*).

Steviol glycosides, extracted from the leaves of *Stevia*, are widely used as a natural sweetener in the food industry because of their high sweetness, low calories, and easy solubility (*Tadhani, Patel & Subhash, 2007*; *Karimi et al., 2019*). China is the center of *Stevia* cultivation, and the cultivation area is continuously increasing due to increasing demand and prices (*Shahnawaz et al., 2021*). This crop is usually grown in continuous cropping

by transplanting cutting seedlings every spring. Many studies have demonstrated that continuous cropping affects soil fertility by influencing soil physicochemical characteristics and microflora (*Wang et al., 2023a*; *Zhang et al., 2022a*; *Zhang et al., 2022b*; *Wang et al., 2023b*). The yield and quality of *Stevia* are dependent on soil physicochemical and microbiome status (*Castrillo et al., 2017*). However, it is unclear how the soil physicochemicals and microbiome change after *Stevia* cultivation. Clarifying these changes will contribute to a more profound understanding of the effects of *Stevia* cultivation on soil physicochemical and microflora and provide guidance for fertilizer management in the rotational cultivation of this crop. Clay and sandy loam soil are more widely distributed in China than sandy soil and are the main soil used for crop cultivation. Clay contains a large number of small particles, which have good fertilizer and water retention properties, but poor air permeability (*Zhao et al., 2015*; *Zheng et al., 2023*). Loam soil has particles of moderate size, with average water and fertilizer retention properties, and moderate air permeability (*Zhao et al., 2015*; *Zheng et al., 2023*). Sandy loam and clay soil are also the dominant soil for *Stevia* cultivation in China. Therefore, in this study, we explored the effects of *Stevia* cultivation on the microbial characteristics of the rhizosphere and non-rhizosphere regions of sandy loam and clay soil using 16S rDNA high-throughput sequencing technology. Some soil physicochemical parameters were also determined. In addition, the correlation between soil physicochemical parameters and bacterial community characteristics was analyzed.

## MATERIALS & METHODS

### Study site and sample collection

The experiments were carried out in Jianggang Town, Dongtai City, Jiangsu Province (120.19E, 32.51N). This location has a subtropical monsoon climate with an annual rainfall of 1,042 mm and an annual average temperature of 14.6 °C. *Stevia* was cultivated in sandy loam and clay soil (Fig. 1), respectively. Table S1 shows the basic physical properties, bulk density, and fertility of the site. The *Stevia* variety "Mao 2" was selected for this study, which is cultivated locally in Jiangsu province. *Stevia* seedlings were transplanted on March 22, 2019, and harvested on August 5 of the same year. The planting density was 15,000 plants per hectare. Before planting, nitrogen, phosphorus, and potassium compound fertilizer (N: $P_2O_5$: $K_2O$ = 15: 15: 15) were used as base fertilizer (45 kg/ha), and urea and potassium dihydrogen phosphate were applied later. Other methods were the same as conventional field management (*Xu et al., 2024*).

In the late stage of *Stevia* maturity, an S-shaped sampling method was used to collect soil samples from each plot as described by (*Xu et al., 2024*), and roots and soil at 0∼20 cm depths were dug out with a shovel. The five samples from five plants were combined and represented one replicate, with a total of three repeats. Soil that was gently shaken off from the roots was collected as bulk soil, and soil that was gently brushed off from the roots with a brush was collected as rhizosphere soil (*Jiang et al., 2020*). The rhizosphere soil and bulk soil samples of the sandy loam and clay soil were denoted as Rs, Bs, Rc, and Bc, respectively. The soil samples were divided into two parts: one part was air-dried to test the physical

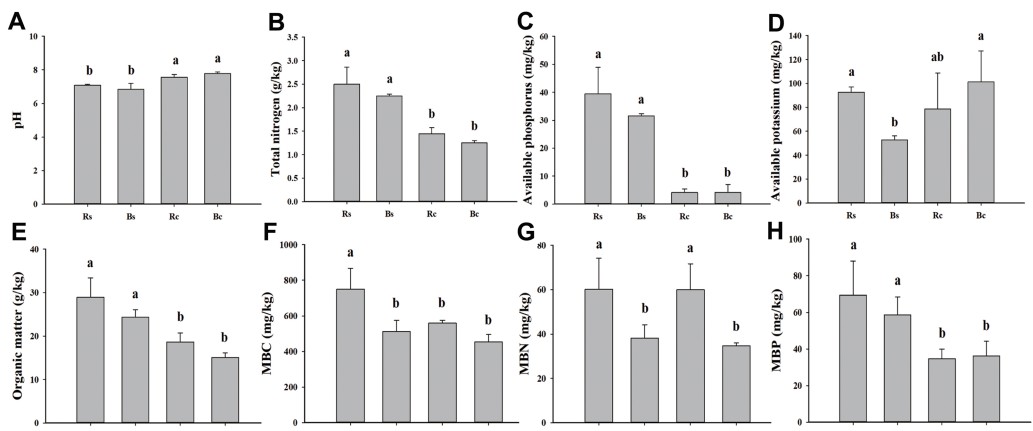

**Figure 1** **A comparison of soil physicochemical properties and microbial biomass of rhizosphere soil and bulk soil in sandy loam soil and clay soil after stevia cultivation.** (A) Soil pH. (B) Total nitrogen. (C) Available phosphorus. (D) Available potassium. (E) Organic matter. (F) Microbial biomass carbon (MBC). (G) Microbial biomass nitrogen (MBN). (H) Microbial biomass phosphorus (MBP). Different letters after the numbers in the figure represent significant differences between samples in the same column ($p < 0.05$). Acronyms: Rs, rhizosphere soil of sandy loam soil; Bs, bulk soil of sandy loam soil; Rc, rhizosphere soil of clay soil; Bc, bulk soil of clay soil.

and chemical properties, while the other part was sieved through a 0.2 mm mesh to test the soil microbial biomass and placed in a refrigerator at −80 °C for DNA extraction.

## Soil physicochemical analysis

The physicochemical properties of the soil and microbial biomass were determined according to the previous methods (*Shen et al., 2015*; *Wang et al., 2011*). Using the value of the soil suspension with a water/soil ratio of 2.5:1 determined using a soil pH value, the potassium dichromate method was used to determine soil organic matter (OM). The flow analyzer (AA3, Seal, Germany) method was used to determine total nitrogen (TN). The sodium bicarbonate-molybdenum antimony colorimetric method was used to determine soil available phosphorus (AP), and the ammonium acetate-flame photometer method was used to determine soil available potassium (AK). The chloroform fumigation-$K_2SO_4$ extraction method was used to determine soil microbial biomass carbon (MBC), microbial biomass nitrogen (MBN), and microbial biomass phosphorus (MBP).

## DNA extraction, amplification, and sequencing

A DNA extraction kit (ALFA-SEQ advanced soil DNA Kit (mCHIP, Inc, Guangzhou, China) was used to extract genomic DNA from the soil samples, and a NanoDrop One was used to detect DNA concentration and purity. Primers 338F (5′-ACTCCTACGGGAGG CAGCA-3′) and 806R (5′-GGACTACHVGGGTWTCTAAT-3′) were used to amplify 16S rDNA V3-V4 sequences. The PCR amplification reaction system consisted of 25 µl of 2X Premix Taq, 1 µl of Primer-F (10 mM), 1 µl of Primer-R (10 mM), 3 µl of DNA (20 ng/µl), and 20 µl of nuclease-free water. The PCR reaction parameters were 94 °C denaturation for 5 min, 94 °C for 30 s, 52 °C for 30 s, and 72 °C for 30 s for a total of 30 cycles; followed

by a 10-min 72 °C extension. GeneTools Analysis Software (Version 4.03.05.0, SynGene, Bangalore, India) was used to compare the concentrations of the PCR products, calculate the required volume of each sample according to the principle of equal quality, and mix the PCR products. The PCR mixture was recovered with an EZNA® Gel Extraction Kit, and the target DNA fragments were recovered with TE buffer. The library was constructed according to the standard procedure of the NEBNext® Ultra™ DNA Library Prep Kit for Illumina®. The libraries were sequenced using Illumina HiSeq2500 platform (Guangdong Meige Gene Technology Co., Ltd., Guangdong, China).

## Processing of sequencing data

Trimmomatic software was used to process the original data to remove low-quality sequences (reads with Ns, reads with a quality value of less than 20, and reads with a filtered sequence of less than 100 bps). The sequence after quality control was spliced by FLASH, the minimum overlap was 10 bps, and the maximum mismatch ratio was 0.1. After splicing the sequence, the sample sequence was segmented using Mothur software according to the barcode and primer information, and then the barcode and primer were removed, resulting in a spliced fragment. High-quality sequences were clustered into operational taxonomic units (OTUs) of 97% similarity using usearch (v10.0.240). After removing chimeras and singletons, the most frequent sequence was selected as the representative sequence of each OTU for subsequent annotation. The species information was obtained by comparing the OTU representative sequence with the Silva database using QIIME. The OTUs that were annotated as chloroplasts and mitochondria, and that could not be annotated to the boundary level were removed, resulting in 2,455–3,681 OTUs per sample.

## Statistical analysis

SAS 9.2 was used for analysis of variance (ANOVA), and the least significant difference (LSD) method was used for multiple comparisons. The significance level was set at $P < 0.05$. Sigmaplot 10.0 was used to illustrate the figures. Based on the relative abundance in the OTU table, the alpha diversity and beta diversity of the samples were analyzed with R software. Redundancy analysis (RDA) of the community structure and soil environmental factors was carried out, and a heatmap was drawn.

# RESULTS

## Soil physicochemical properties and microbial biomass

The basic physicochemical properties and microbial biomass of rhizosphere soil and bulk soil in sandy loam and clay soil are listed in Fig. 1. The pH value of the sandy loam soil was significantly lower, and the rhizosphere soil and bulk soil were 6.35% and 12.08% lower than the clay soil, respectively. The MBP of the rhizosphere soil and bulk soil in the sandy loam soil was 50.04% and 38.37% higher than in the clay soil, respectively. The AK, MBC, and MBN of the sandy loam soil were significantly higher than the clay soil. No significant differences were observed among the other indicators between the rhizosphere soil and bulk soil, except for the MBC and MBN of the sandy loam soil and the MBN of the clay soil.
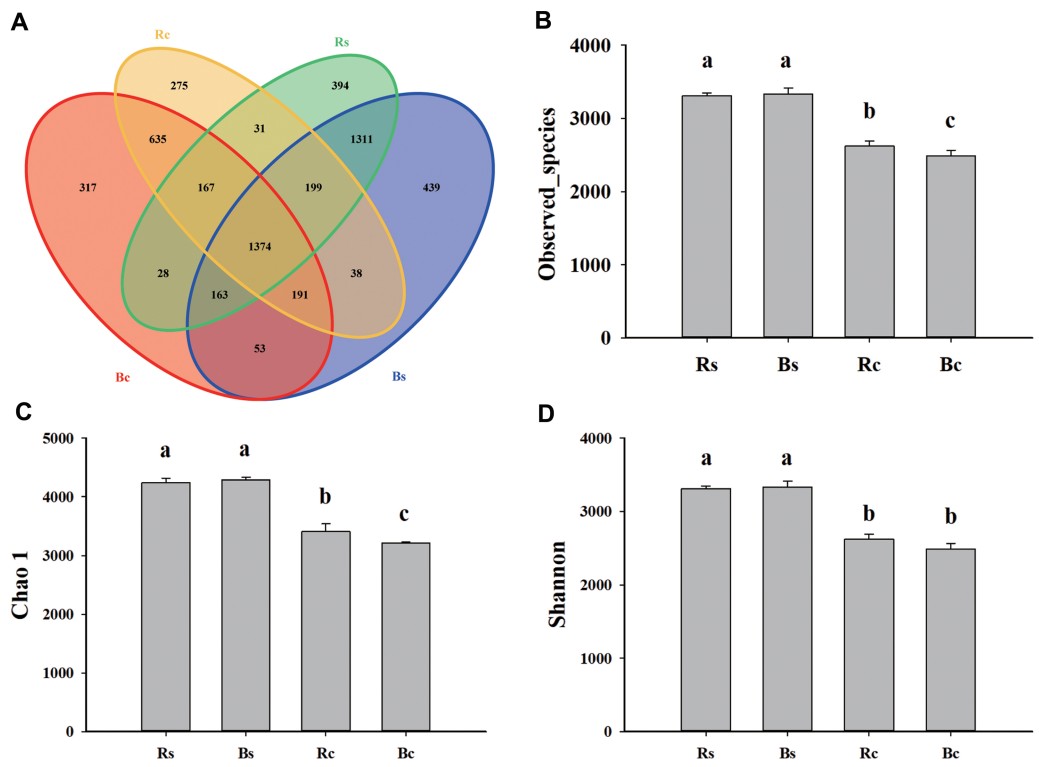

**Figure 2 The Venn diagram and alpha diversity index of the soil rhizosphere bacterial community were rendered by 16S rRNA sequencing.** (A) The Venn diagram for sandy loam soil and clay soil based on the OTU shared at 0.03 dissimilarity distances after removing singletons. (B) The observed species index. (C) The Chao 1 index. (D) The Shannon index. Acronyms: Rs, rhizosphere soil of sandy loam soil; Bs, bulk soil of sandy loam soil; Rc, rhizosphere soil of clay soil; Bc, bulk soil of clay soil.

## Comparison of bacterial alpha and beta diversity between sandy loam soil and clay soil

A total of 1,159,961 quality sequences and 1,025,361 valid sequences were obtained from 12 samples, resulting in a total of 5,937 OTUs (Table S2). The rarefaction curves and Shannon-Wiener index curves tended to become flat with the increase in the number of sequences, which indicated that the amount of sequencing data was reasonable (Fig. S1). There were 7,435 and 5,838 at the 97% similarity level shared among systems in sandy loam soil and clay soil, respectively. For both rhizosphere soil and bulk soil, higher OTU numbers were observed in sandy loam soil (3,667 and 3,768) than in clay soil (2,910 and 2,928). The bulk soil of sandy loam soil had the fewest unique OTUs (439), and the rhizosphere soil of clay soil had the fewest unique OTUs (275). In addition, the OTUs numbers that were commonly detected between rhizosphere soil and bulk soil accounted for 37.47% and 36.46% in sandy loam soil, respectively, and 47.22% and 46.93% in clay soil, respectively (Fig. 2A). Likewise, significantly ($P < 0.05$) higher observed species, Chao 1 and Shannon indices, were observed in sandy loam soil than in clay soil (Figs. 2B–2D). Among them, no difference was observed between rhizosphere soil and bulk soil in the sandy loam soil, while observed species index and chao 1 index of clay soil were significantly higher in

**Table 1  Pearson's correlation coefficients between bacterial diversity indices and soil physicochemical properties.**

| indices | pH | OM | TN | AP | AK | MBC | MBN | MBP |
|---|---|---|---|---|---|---|---|---|
| Observed species | −0.84826** | 0.8532d** | 0.91640** | 0.94318** | −0.46330 | 0.52670 | 0.10973 | 0.82008** |
| Chao 1 | −0.86262** | 0.83208** | 0.91127** | 0.93399** | −0.47284 | 0.51083 | 0.13630 | 0.81334** |
| Shannon | −0.81302** | 0.82416** | 0.83626** | 0.87913** | −0.54994 | 0.50517 | 0.13395 | 0.76865** |

**Notes.**

pH, soil pH; OM, organic matter; TN, total nitrogen; AP, available phosphors; AK, available potassium; MBC, microbial biomass carbon; MBN, microbial biomass nitrogen; MBP, microbial biomass phosphorus; Rs, rhizosphere soil of sandy loam soil; Bs, bulk soil of sandy loam soil; Rc, rhizosphere soil of clay soil; Bc, bulk soil of clay soil.

** means $p < 0.01$.

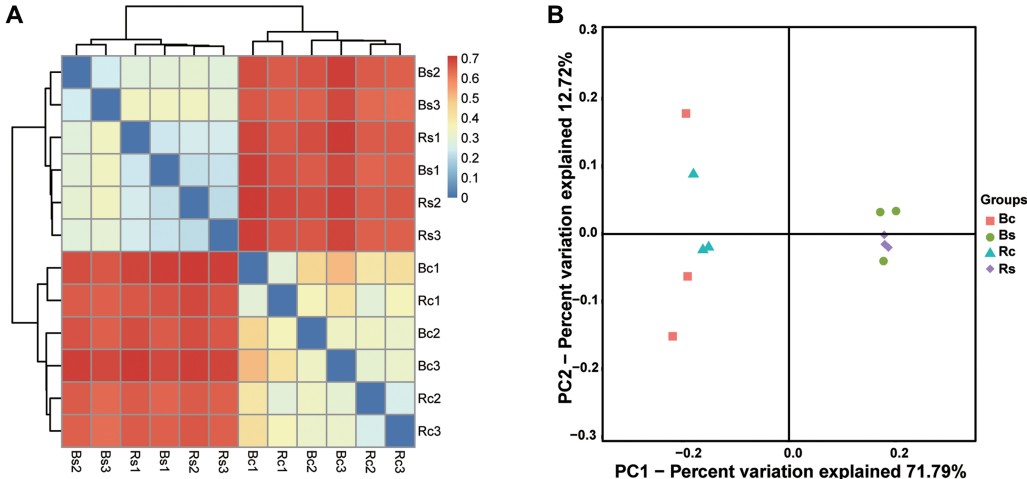

**Figure 3  The bacterial beta diversity.** (A) Bray_Curtis cluster; (B) weighted_Unifrac_PCoA. Acronyms: Rs, rhizosphere soil of sandy soil; Bs, bulk soil of sandy loam soil; Rc, rhizosphere soil of clay soil; Bc, bulk soil of clay soil.

the rhizosphere soil than in the bulk soil (Figs. 2B–2D). Pearson's correlation analysis further showed that the observed species, Chao 1 and Shannon indices, were significantly negatively correlated with soil pH but significantly positively correlated with OM, TN, AP, and MBP ($P < 0.01$) (Table 1).

The clustering results obviously showed that the bacterial communities in two soil textures had a high dissimilarity, which was consistent with the difference in soil physicochemical properties (Fig. 3A). Principal coordinate analysis (PCoA) based on the OTUs composition similarities showed that the first two dimensions explained 59.58% of the total variance in the soil bacterial community. The principal coordinate components of the bacterial community in the sandy loam soil were clearly distinguished from the clay soil on one axis, and the two soil samples were distributed far apart, indicating that the bacterial community structure of the two soil samples was quite different. Moreover, the sandy loam soil samples clustered together, while the clay soil samples were relatively dispersed (Fig. 3B). The result of the ADOSIM analysis for the sandy loam soil and clay soil also showed differences in the structure of the soil bacterial community ($R^2 = 0.74$, $p = 0.004$), which was consistent with the above alpha and beta analysis results.

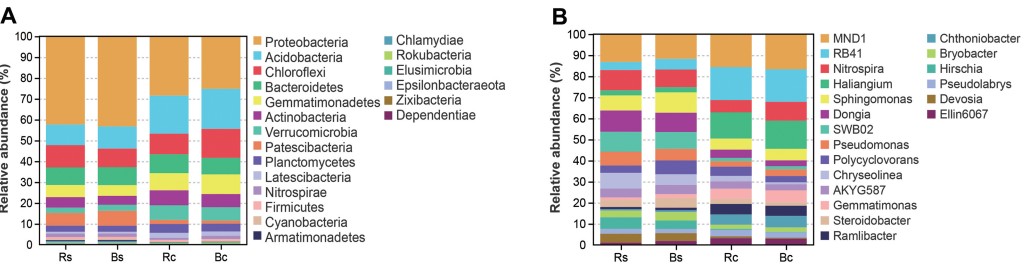

**Figure 4** **The relative abundance of top 20 bacterial phyla (A) and genera (B) in the rhizosphere soil and bulk soil of sandy loam soil and clay soil.** Acronyms: Rs, rhizosphere soil of sandy loam soil; Bs, bulk soil of sandy loam soil; Rc, rhizosphere soil of clay soil; Bc, bulk soil of clay soil.

## Comparison of bacterial community composition between sandy loam soil and clay soil

Figure 4 shows the relative abundance distribution of bacterial composition in each group. Obviously, the two soil textures were rich in different phyla and genera of bacteria. The dominant phyla were Proteobacteria, Acidobacteria, Chloroflexi, Bacteroidetes and Gemmatimonadetes, and these phyla accounted for over 75% of the bacterial community, among which the highest relative abundance (25.13%~42.95%) was calculated for Proteobacteria (Fig. 4A). Among the top 10 phyla (>0.5%), the relative abundance of Proteobacteria, Acidobacteria, Gemmatimonadetes, Verrucomicrobia, Patescibacteria and Latescibacteria showed significant differences among the four soil samples ($P < 0.05$). By contrast, the relative abundance of Proteobacteria and Patescibacteria in the rhizosphere soil or bulk soil of the sandy loam was significantly higher than in the clay soil ($P < 0.05$), while the relative abundance of Acidobacteria, Gemmatimonadetes, Verrucomicrobia, and Latescibacteria in the sandy loam soil was significantly lower than in the clay soil ($P < 0.05$).

Figure 4B further illustrates the distribution of the dominant bacterial genus (>1.5%) in the two soil textures. In the top 20 genera, 11 genera showed a significant difference between the two soil textures ($P < 0.05$). Among them, only *Hirschia* had a significant difference between rhizosphere soil and bulk soil. The relative abundance of *RB41*, *Haliangium* and *Ramlibacter* in sandy loam soil was significantly lower than in clay soil, and the relative abundance of *Dongia*, *SWB02*, *Chryseolinea*, *Bryobacter Hirschia* and *Devosia* in sandy loam soil was significantly higher than in clay soil ($P < 0.01$). Figure S2 shows in detail the comparison of Proteobacteria and its five differential genera under different treatments.

## Relationship between bacterial community structure and soil properties

Redundancy analysis (RDA) was applied to analyze the relationships between the bacterial community and the soil's physicochemical properties (Fig. 5A). RDA explained 95.21% of the total variation in the two soil textures. Soil pH, OM, TN, AP, and MBP were the significant factors that explained differences in relative abundance at the phylum level ($P < 0.01$) (Table S3). Thereafter, we performed the correlation heatmap analysis to obtain the relationship between soil physicochemical properties and specific bacterial
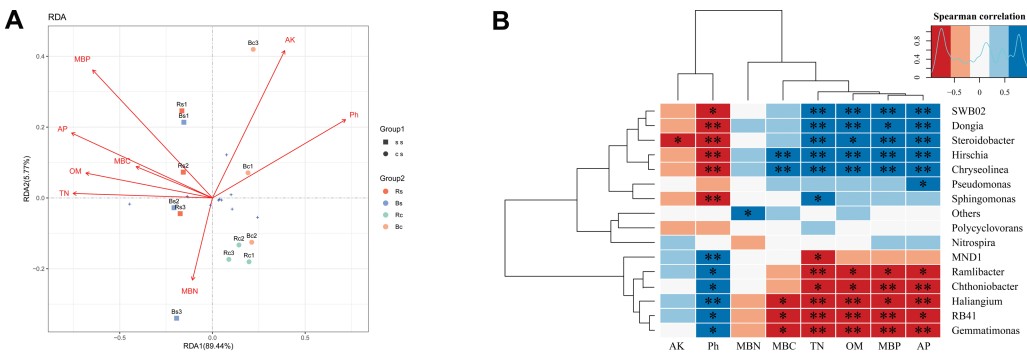

**Figure 5** **A correlation analysis between soil microbiome composition and soil environmental variables.** (A) Redundancy analysis (RDA) between bacterial composition at the phylum level and soil physicochemical properties. The arrows represent different environmental factors. The longer the ray, the greater the impact of the environmental factor. When the angle between environmental factors is an acute angle, the two environmental factors are positively correlated, and when the angle is an obtuse angle, they are negatively correlated. (B) Correlations among dominant bacterial communities at the genus level and physicochemical properties. Red represents a negative correlation, and blue represents positive correlation. Acronyms: pH, soil pH; OM, organic matter; TN, total nitrogen; AP, available phosphors; AK, available potassium; MBC, microbial biomass carbon; MBN, microbial biomass nitrogen; MBP, microbial biomass phosphorus. Acronyms: Rs, rhizosphere soil of sandy loam soil; Bs, bulk soil of sandy loam soil; Rc, rhizosphere soil of clay soil; Bc, bulk soil of clay soil; ss, sandy loam soil; cs, clay soil. * means $p < 0.05$, ** means $p < 0.01$.

genus (Fig. 5B). Soil pH was significantly correlated with other known species, except for *Pseudomonas*, *Polycyclovorans* and *Nitrospira*. The relative abundance of *SWB02*, *Dongia*, *Steroidobacter*, *Hirschia*, and *Chryseolinea* showed a significantly negative correlation with soil pH and a significantly positive correlation with TN, OM, AP and MBP. The relative abundance of *Ramlibacter*, *Chthoniobacter*, *Haliangium*, *RB41*, and *Gemmatimonas* showed a significantly positive correlation with soil pH and a significantly negative correlation with TN, OM, AP, and MBP.

## DISCUSSION

Microbial biomass is not only a dominant factor in the transformation of organic matter in the soil, but also a source and sink of soil nutrients. It reflects, to some extent, the fertility of the soil (*Xun et al., 2015*; *Chen et al., 2020*). It has been demonstrated that sandy loam soil has a higher organic matter conversion rate in the rhizosphere and bulk soil than in clay soil, resulting in more accumulation of soil organic matter more abundant microbial inhabitants (*Sokol et al., 2022*; *Wang et al., 2021*). In this study, when *Stevia* was harvested, the sandy loam soil was found to have significantly greater organic matter content in the rhizosphere and non-rhizosphere than the clay soil. Our study also found that there was a significantly positive correlation between alpha diversity indices and organic matter content. In addition, it showed that the rhizosphere MBC and MBP of the sandy loam and its non-rhizosphere MBP were significantly greater than those of the clay. It can be inferred that compared with clay soil, the sandy loam soil cultivated with *Stevia* has a larger microbial biomass and richer microbial population due to its rich organic matter content,

which is beneficial for improving their fertility and environment. This also supports the above viewpoint that sandy loam soil has a richer microbial biomass and population derived from higher organic matter conversion. It implies that sandy loam soils have a higher organic matter content and microbial biomass than clay soil after *Stevia* cultivation.

This study revealed that the pH of rhizosphere and non-rhizosphere soil in sandy loam soil was significantly lower than that of clay soil and that there was a significant negative correlation between soil pH and alpha diversity indices such as observed species, chao 1 and shannon. Previous studies have confirmed that soil pH exerts a strong influence on the structure of soil microbial communities by regulating soil nutrient availability (*Shrivastav et al., 2020*; *Zhang et al., 2020a*; *Zhang et al., 2020b*). Soil pH can partly explain and predict the diversity and richness of soil bacterial communities found in different ecosystems (*Fierer & Jackson, 2006*; *Lauber et al., 2009*). Soil pH is recognized as one of the key factors influencing soil microbial communities and is usually negatively correlated with community abundance (*Tripathi et al., 2012*; *Zhang et al., 2016*; *Tripathi et al., 2018*; *Yang, Dou & An, 2018*). *Stevia* grown on acidic soil has better growth status and higher biomass than *Stevia* grown on alkaline soil (*Zaman, 2015*). This study demonstrates that *Stevia* grown on low pH sandy loam soil has better growth than on high pH clay soil, and low pH sandy loam soil has a richer beneficial microbial community than high pH clay soil. Therefore, it suggests that these beneficial microorganisms may promote the growth and biomass of *Stevia*.

It has been proven that sandy loam soil contains more diverse microbial species than clay soil after tobacco (*Nicotiana tabacum*) cultivation because that the former has a higher total nitrogen and available phosphorus content than the latter (*Li et al., 2012*). Our study found that the TN and AP contents in the rhizosphere and non-rhizosphere of sandy loam soil were substantially higher than in those of clay soil following *Stevia* cultivation, and that these two soil parameters were significantly positively associated with alpha diversity indices. The results of beta diversity also indicated that the two soil textures had distinct dominant bacterial phyla and genera after *Stevia* cultivation. These results suggest that the sandy loam soil held higher total nitrogen and available phosphorus contents and more abundant microbial species after *Stevia* cultivation.

Our study indicated that soil texture has significant effects on soil bacterial community structure after cultivation. At the phylum level, Proteobacteria were more abundant in the sandy loam soil than the clay soil; however, Acidobacteria have a greater population in clay soil than in sandy loam soil. It has been reported that the application of organic matter and nitrogen in soil can promote the growth and population increase of Proteobacteria (*Cesarano et al., 2017*; *Dai et al., 2018*). In current research, sandy loam soil contains higher levels of organic matter and nitrogen elements than clay soil. This suggests that compared to clay soil, the higher organic matter and nitrogen content in sandy loam soil leads to a more Proteobacteria population. At the genus level, *Haliangium, RB41,* and *Ramlibacter* were richer in clay soil than in sandy loam soil, but *Dongia, Devosia, Chryseolinea,* and *Bryobacter* were more abundant in sandy loam soil than in clay soil. Proteobacteria improves soil fertilizer by increasing soil organic matter (*Dai et al., 2018*) and promotes plant growth by establishing a mutualistic association with plant roots. (*García-Serquén et al., 2024*),

but Acidobacteria belong to the class of oligotrophic bacteria (*Pascual et al., 2015*; *Liu et al., 2017*). *Haliangium, RB41,* and *Ramlibacter* are related to the lack of soil phosphorus and phytopathogens (*Ma et al., 2018*; *Chen et al., 2022*). *Dongia, Devosia, Chryseolinea,* and *Bryobacter* are associated with higher available phosphorus content, bioremediation potential, and decomposition ability (*Lu et al., 2020*; *Pertile et al., 2021*; *Zhang et al., 2022a*; *Zhang et al., 2022b*). These results suggest that after *Stevia* cultivation, sandy loam soil has more beneficial microbial communities, while clay soil has more unfavorable microbial communities.

In practical operations, *Stevia* is often cultivated in rotations. Soil characteristics and fertility after the cultivation can influence fertilization management in the next cultivation of this crop. In this study, different soil textures had distinct physicochemical and microbiological parameters after *Stevia* cultivation, which implies that different fertilization strategies should be applied to various soil textures during *Stevia* rotation to improve soil fertility for the upgradation of yield and quality. For example, in the next *Stevia* cultivation, more fertilizers such as organic, nitrogen, phosphorus, and mycorrhizal fertilizers should be applied in the clay soil than in the sandy loam soil to improve the physicochemical parameters and microbial abundance for the increase in fertility. Although our study provides a better understanding of the soil physicochemical and microbial characteristics of sandy loam and clay soil following *Stevia* cultivation, more work needs to be done in the future. Changes in parameters of other soil textures after *Stevia* cultivation should be studied. Soil microbial metabolome and proteome after *Stevia* cultivation also need to be examined. In addition, how and why different soil textures affect *Stevia* yield and quality should be studied. Currently, we are carrying out some work, such as proteomic and metabolomic studies of soil microorganisms during *Stevia* rotations.

## CONCLUSIONS

The effects of a crop cultivation on soil determines the land management and fertilization strategies for the rotation of the crop and the subsequent planting of other crops. The cultivation of different crops may have various impacts on the physiological, biochemical, and microbial characteristics of different texture soils. Stevia is a widely cultivated crop on sandy loam and clayey soil. Our research found that sandy loam soil after cultivating stevia has better fertility than clayey soil after cultivating stevia. Compared with the clay soil after cultivating stevia, sandy loam soil after cultivating stevia has a higher content of organic matter, nitrogen, potassium, and more beneficial microorganisms. Stevia growing in sandy loam soil can achieve better growth and higher biomass than in clay soil. This will help to better understand the mechanisms of soil fertility changes in different textures during the rotation of stevia, and provide reference for soil and fertilizer management. Although we have conducted research on the effects of stevia cultivation on the characteristics of sandy loam and clay soil, it will also provide a contribution to understanding the effects of other crop cultivation on soil characteristics.

## ACKNOWLEDGEMENTS

We would like to express our sincere gratitude to Dr. Jiang Shangtao for his enthusiastic assistance during the experimental material collection process.

### Funding

This work was supported by the Key Scientific and Technological Projects in Henan Province (No. 242102111082), the Henan University of Science and Technology High level Talent Introduction Program Project (No. 103020224001/003) and the Research on Sustainable Utilization of Chinese Stevia Producing Area (Dongtai) (No. HMQT21028). The funders had no role in study design, data collection and analysis, decision to publish, or preparation of the manuscript.

### Grant Disclosures

The following grant information was disclosed by the authors:
Key Scientific and Technological Projects in Henan Province: 242102111082.
Henan University of Science and Technology High level Talent Introduction Program Project: 103020224001/003.
Research on Sustainable Utilization of Chinese Stevia Producing Area: HMQT21028.

### Competing Interests

The authors declare there are no competing interests.

### Author Contributions

- Xinjuan Xu conceived and designed the experiments, performed the experiments, analyzed the data, prepared figures and/or tables, authored or reviewed drafts of the article, and approved the final draft.
- Qingyun Luo conceived and designed the experiments, performed the experiments, prepared figures and/or tables, authored or reviewed drafts of the article, and approved the final draft.
- Ningnan Zhang performed the experiments, analyzed the data, prepared figures and/or tables, authored or reviewed drafts of the article, and approved the final draft.
- Yingxia Wu performed the experiments, prepared figures and/or tables, and approved the final draft.
- Qichao Wei analyzed the data, prepared figures and/or tables, and approved the final draft.
- Zhongwen Huang analyzed the data, prepared figures and/or tables, authored or reviewed drafts of the article, and approved the final draft.
- Caixia Dong conceived and designed the experiments, authored or reviewed drafts of the article, and approved the final draft.

## DNA Deposition

The following information was supplied regarding the deposition of DNA sequences:

Soil bacterial community after stevia growth in different soil texture are available in the OMIX database: OMIX001064.

## Data Availability

The raw measurements are available in the Supplementary Files.

## Supplemental Information

Supplemental information for this article can be found online at http://dx.doi.org/10.7717/peerj.18010#supplemental-information.

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
