# Peer review of "Sandy loam soil maintains better physicochemical parameters and more abundant beneficial microbiomes than clay soil in Stevia rebaudiana cultivation"

_PeerJ, doi:10.7717/peerj.18010_

## Round 0.1 · original submission · Minor Revisions

We have received three reviews with some relevant comments. I believe the remarks are rather technical ones. Please check the PDF file attached and the comments kindly provided by reviewer #3.

Reviewer 1 ·

Basic reporting

This study compared the soil physicochemical properties and microbial communities between sandy loam and clay soils under Stevia cultivation.

The manuscript is clearly written in professional, unambiguous English. The literature is well referenced, providing sufficient context. The structure conforms to standards, with relevant figures, tables, and raw data provided.

Experimental design

The research question is well-defined, and the methods are described in sufficient detail to allow replication. But some minor results should be clarified.

1. Line 100-111: please cite paper to show the reason why S-shaped sampling method was used.

2. Line 134: please explain why to choose V3-V4 region as a representative of 16S.

3. Line 157: what parameters are used for QIIME?

4. Line 161: please spell out the full name of ANOVA

Validity of the findings

The findings appear valid based on the rigorous investigation and extensive data set. However, some improvements could enhance the statistical analysis and interpretation.

Table 1: Significant correlations are shown between soil properties and alpha diversity indices, but these results are not discussed or interpreted in the text. The authors should expand on the implications of these correlations in the Discussion section.

Figure3B: Please explain why Bc and Rc are not separated clearly?

Overall, this is a solid manuscript with valid and interesting findings. Addressing the few areas noted above would further strengthen the publication.

Reviewer 2 ·

Basic reporting

The manuscript entitled 'Sandy loam soil maintains better physicochemical parameters and more abundant beneûcial microbiomes than clay soil in Stevia rebaudiana cultivation' mainly investigates the effects of stevia cultivation on soil microorganisms and soil physiological and biochemical characteristics in sandy loam and clay soil. It is beneficial for people to better understand the impact of stevia rotation on different types of soil and may guide the fertilization of stevia cultivation. Comments for revision are included below.
General Comment
The manuscript is written well in terms of the English language, however, kindly proofread again and try to improve some sentences containing grammatical lapses.
Specific comments
Lines 49, 93, 252 etc. Remove the s after soils.
Lines 49-50, 53, 55 etc. When citing references, use parentheses instead of square brackets.
Line 50 What type of soil does soil refer to?
Lines 50-52 I cannot understand the meaning conveyed by this sentence, please rewrite it.
Line 55, 73 etc. When mentioning a certain plant for the first time, its Latin name should be indicated, such as maze (Zea mays).
Lines 81-83 Merge these two sentences into one sentence and delete “in this country”.
Line 88, 91, 92 etc. As a genus name, the first letter of Stevia should be capitalized and written in italics.
Lines 91. Is there a missing soil after Sandy loam and clay?
Line 104 Is should be changed to was.
Line 123 Please provide the model and production company of the flow analyzer.
Line 162, 189, 196 etc. Please change the letter p to uppercase.
Line 239, 242 SWB02 and RB41 should be written in italics.
Lines 261-263 This sentence is too long, please express it in two sentences to make it easier for readers to understand.
Line 304 Should types be changed to textures?
Line 304 Improve and improvement are duplicated, please replace them with others.

Experimental design

no comment

Validity of the findings

no comment

Additional comments

no comment

Reviewer 3 ·

Basic reporting

The MS is clear with updated references. The MS is well written with good quality figures.

Experimental design

The methodological approaches are relevant to the study design. The research question is well-defined, and has been answered by the authors with good approaches. even though some improvement still needed

Validity of the findings

All data has been statistically analyzed. conclusion still need to be written in more detail.

Additional comments

The paper aims to evaluate the effect of stevia cultivation on physicochemical and microbiological parameters. Physicochemical analysis, biochemical and microbiome approach have been carried out. Authors presented good and comprehensive data. However, I believe that some issues should be improved before its consideration to publication.

1. Abstract are well written and compact, consisting of background, objective, methodological approach, results, conclusions and prespective.
2. Introduction section is compact and contain important related to the study. However, author should provide more information about soil type used in stevia cultivation, rather than use maize case in this study (Line 55).
3. Soil microbiome plays important role in stevia cultiavation (L60-67). Authors are asked to give some examples of microbial communities often found in rhizospheric soil of stevia cultivation. Current reference such as https://doi.org/10.1016/j.indcrop.2022.115434 and https://doi.org/10.1007/s00374-013-0777-7 could help author providing the information.

4. Sandy loams and clay soils are two types of soils used for stevia cultivation. Authors are asked to give information about it. Why these types of soils are currently used for stevia cultivation, as well as their nature (preferred physcochemical characters). What do stevia need to maintain their good growth rate with high yields.
5. Lines 79-84 provide information about the importance of stevia cultivation. High demand for the plant necessitates good soil environment and good quality of plant. Author should provide information how stevia could be propagated? Both using conventional method or plant tissue culture. A recent publication such as : https://doi.org/10.1186/s13104-024-06703-0 might help author to provide this information.

6. Legend in the Figure 1 should be corrected. Phrase “Different letters after the numbers in the table represent significant….”. It is figure/graph, NOT table!
7. Please give table S1 title (also others). Authors should provide the abbreviation of all information contained in the tables/figures!
8. Add relevant reference to “Other methods were the same as conventional field management” (in Line 109).
9. Line 251-253 : in what type of plants? is there any explanation that type of plants might also influence the physicochemical characteristics of the soils?
10. In line 261: “…there was a significantly positive correlation between microbial species and organic matter content and that the rhizosphere MBC and MBP of the sandy loam and its non-rhizosphere MBP were significantly greater than those of the clay….”. The authors should give additional explanation about it!
11. Figure 2b : observed species between rhizosphere and bulk soil (sandy loams) are not significanly different. Why was it?
12. Line 266 : “pH of rhizosphere and non-rhizosphere soils in sandy loam soil was significantly lower than that of clay soils”. Authors should explain how pH is important for stevia not only for the bacteria? do stevia prefer lower pH or higher pH? Explain why? Make correlation between favorable microbes with plant growth (stevia)!
13. Line 287 : why Proteobacteria is abundant in sandy loam than clay? How proteobacteria could improve soil fertilizer and crop yield (line 292)
14. Conclusions should be made in more detailed and open perspectives should be written.

Annotated reviews are not available for download in order to protect the identity of reviewers who chose to remain anonymous.

---

## Round 0.2 · accepted · Accept

Thanks for the manuscript update and detailed remarks to the reviewers. There are no more critical remarks.

Reviewer 3 ·

Basic reporting

The revised article is now clear and demonstrates good written style with updated references.

Experimental design

The material and methods were presented in a good structure and described sufficient information of the experiments.

Validity of the findings

the conclusions are well stated with perspective of the future study.

Additional comments

The authors have made a good improvement in the article. Therefore, I recommend to accept this manuscript.